# The Impact of Visits to Dryland Forests on Environmental Outlook: Results from a National Survey

**Alon Tal** [1,*] 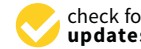 **and Miriam Billig** [2]

1    Department of Public Policy, Tel Aviv University, Tel Aviv-Yafo 6997801, Israel
2    Department of Sociology and Anthropology, Ariel University, Eastern R&D Center, Ariel 40700, Israel; billigm@ariel.ac.il
*    Correspondence: alontal@tau.ac.il

**Abstract:** The effect of visits to the country's forests on environmental perspectives and commitments was assessed in a national survey of the Israeli public. As a highly urbanized country, visits to the country's dryland forests constitute an important national pastime across ethnic lines. We evaluated the impact that forest visitation had on the attitudes and perspectives of the visitors. A strong correlation was found between the frequency of visits to forests by individuals and a range of pro-environmental and pro-conservation sentiments. Of particular interest was the response of Israel's Arab citizens. Not only do Israeli Arabs visit forests more frequently, but they also support environmental policy positions more avidly than do Jewish citizens. The article argues that increasing accessibility to forests and natural sites by expanding public transportation lines should constitute an important component in conservation strategies.

**Keywords:** forests; Israel; visits; effect; environmental awareness; dryland forests; Arab; Ultra-Orthodox citizens

## 1. Introduction

"And into the forest I go, to lose my mind and find my soul…"
John Muir

A "walk in the woods" produces myriad benefits for people [1,2]. Forests are also places where transformational, personal experiences take place. Indeed, there is increasing interest in the role of forest visitations and forestry interventions in fostering environmental values and sustainability behavior [3]. While most people visit forests in a recreational context, the direct interaction with the natural world not only offers an important opportunity for observation, contemplation, inspiration, and learning but is also a transformative experience psychologically and ideologically [4].

The academic literature is replete with studies about the restorative emotional effects of forest visits [5–7]. There are also many well-documented physiological benefits of visiting woodlands [8], notably: cardiological improvements [9], improved immune function, and reduced hypertension [10]. Studies in Japan, for example, indicate that "forest bathing" experiences or brief trips to the nearby woods lead to a statistically significant strengthening of the human immune system [11,12]. A recent meta-analysis of empirical research concluded that one underappreciated "ecosystem" service that humans receive from interactions with nature is improved mental health. The authors argue that increasing routine contact with nature sites could be an important step for improving "health equity" [13].

Bringing school children to forests and allowing for natural interactions has been associated with increasing their confidence, social skills, language and communication, motivation, concentration,

physical skills, and knowledge and understanding [14,15]. For environmental educators who worry about inculcating environmental values, perpetuating conservation ethics, and fostering environmental behavior, forest recreation and visitation are considered excellent potential means for inspiring younger and older generations alike [16,17]. Indeed, as the world's population becomes increasingly urban, the importance of bringing children to the outdoors to learn about the natural environment is increasingly emphasized [18]. Research confirms that the frequency of visits to forested areas [19], "appreciative activities" [20], as well as knowledge about plants [21], are all associated with pro-environmental behavior and concern for the environment.

However, all forests are not alike. It is not clear whether newly planted forests or woodlands designed to survive in dryland environments offer the same psychological and spiritual benefits and educational opportunities as conventional, lush, old-growth stands. Neither is it clear whether political controversy surrounding a forest will affect the educational and psychological experience of the visitor. The idiosyncratic nature of Israeli forests offers an opportunity to explore these questions.

We conducted a national survey of Israelis to evaluate their feelings towards local forests and to assess the impact of forest visitation and recreation on local attitudes towards forests and forest management. The research asks whether differences in the environmental perspective of Israelis can be attributed to the frequency of visits to local forests. The study also asks whether there are significant differences between the impact of visits among the country's diverse ethnic and religious communities. Results are largely consistent with the edifying effects of visiting forests found around the world. We begin our article with a brief review of Israel's experience in forestry, highlighting the features which make the country's woodlands unique. We then detail the methods we used to collect data about Israeli attitudes and experience with regards to local forests. We then present the survey results descriptively and graphically. In the discussion and concluding sections, the implications of the findings with regards to the impact of forest visitation on environmental attitudes are considered along with recommendations for policymakers.

## 2. The Woodlands of Israel

The Hebrew word for *forest* (Ya'ar) appears seventy-one times in the Bible. The word may be ancient, but most forests in Israel are not. Practically none of the local Mediterranean stands conform to conventional, Euro–American stereotypes of rich, lush, and diverse woodlands. The definitive Hebrew dictionary defines a *forest* as "a large area of land where different non-fruit trees grow". The trees growing naturally in the forests of Israel are mainly pine, oak, and terebinth trees [22]. These drought-resistant species must compete for limited rainfall in the country's semi-arid and arid climates. Accordingly, official planting protocols call for considerable distances between saplings, frequently with placement as dispersed as 200 trees per hectare to allow the trees to flourish despite the intermittent rains [23].

There are three reasons why Israel's forests can be considered unique: they tend to be young; they are often linked to political controversies, and, for the most part, are located in drylands. With the exception of some isolated patches of "old growth" stands, Israeli forests are generally young, recently-planted additions to the ancient landscape. During the millennia that transpired before the twentieth century, some 97% of the country's natural woodlands were destroyed by the exploitive occupations of various empires, pervasive poverty, rampant overgrazing, and steady deforestation [24]. During the British Mandate in Palestine during the first half of the twentieth century, as part of an ingenuous impulse to improve the degraded landscape of the "Holy Land", afforestation activities were prioritized by the colonial government [25]. In retrospect, these efforts met with limited success, with only 5400 hectares of afforested lands surviving after the Mandate's thirty-year regime. This can be attributed largely to intentional vandalism and intermittent conflict with the local Arab population. They perceived the new, highly-regulated woodlands as expressions of foreign oppression, and as an affront to traditional pastoral practices, as well as a usurpation of public lands [26].

With the establishment of the State of Israel in 1948, forestry was promoted as a top policy priority by the government [27]. Planting initiatives were framed as an important employment and patriotic undertaking. The country's founding Prime Minister, David Ben Gurion, set an impossibly high objective: foresting 25% of the arid countryside. In a speech to the country's first Parliament, he described the "wrapping" of the land's mountains and slopes with trees as a blessing for future generations [28]. Today, some 11.5% of Israel's countryside is indeed covered with forest [29]. But with their relatively high conifer presence and reduced undergrowth, these new wooded areas typically look very different than the country's original Mediterranean forests. Figure 1 offers a comparison of the limited tree stands existing during the first half of the twentieth century and Israel's present woodlands.

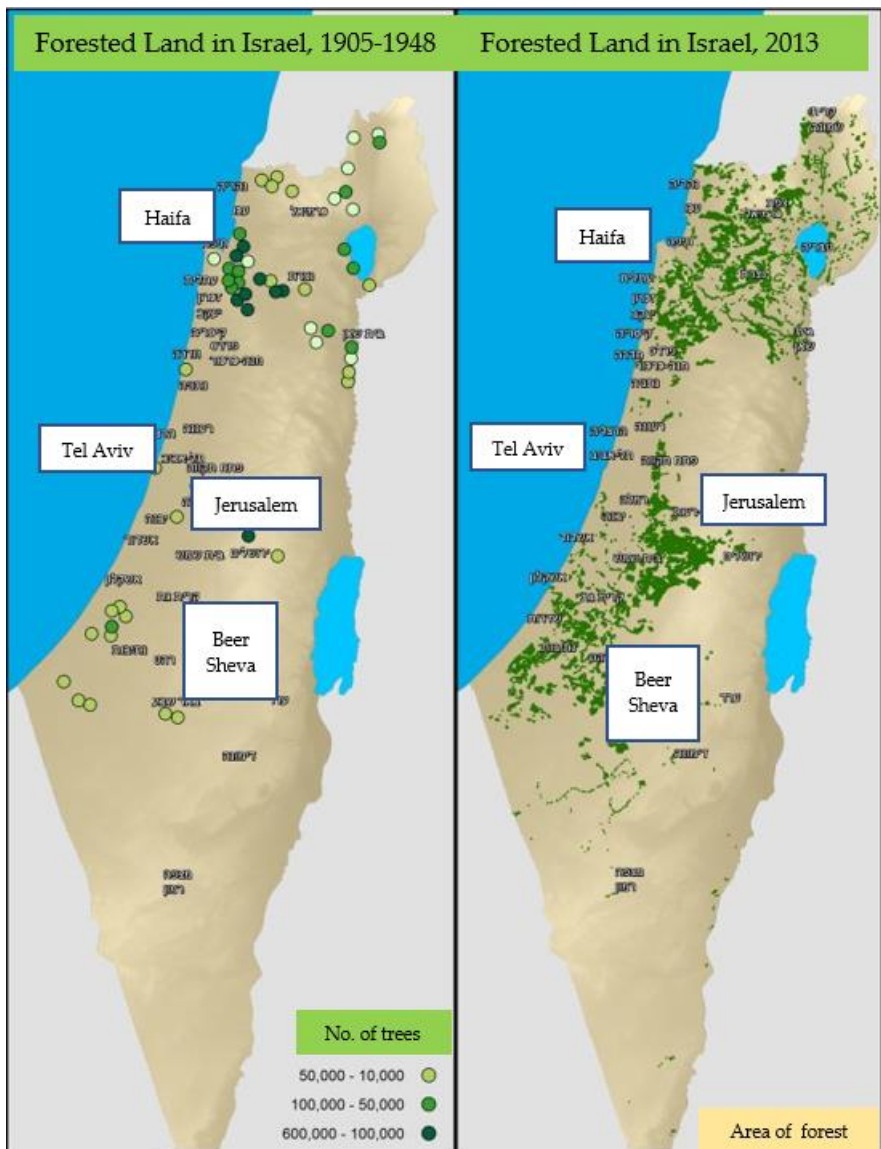

**Figure 1.** Forest expansion in Israel, 1905 to 2013. *Source: DESHE, Israel Open Space Institute.*

Second, Israel's forests have played a meaningful role in local political dynamics. While afforestation has become politicized in many places [30,31], in Israel, the symbolic role of *forest* restoration as part of the Jewish people's national *political* restoration has been particularly conspicuous. Tree planting came to be considered, by some, as a fundamentally political act [32]. Israel's national forestry service is, in fact, a non-government, public corporation—the *Jewish National*

*Fund*, which is typically referred to by its Hebrew acronym—the KKL. The corporation itself is political because appointment to the KKL directorate is based on affiliation with Zionist political parties.

From its inception over a century ago, the company and its afforestation efforts have served as a key implementation arm of the Zionist movement. Immediately after the War of Independence, several forests were intentionally planted on the ruins of abandoned Palestinian villages, presumably to ensure that the inhabitants would not return. This produced an enmity between the KKL and Israel's Arab citizens that lasted for decades. Indeed, many of Israel's forest fires can be traced to politically-motivated arson by alienated Arab citizens [33]. In 1961, the Israeli government signed a covenant with the KKL empowering the company to oversee the replanting and afforestation of forests nationwide, and it continues to do so until the present.

Finally, Israel's forests are almost uniformly located on drylands, where annual precipitation does not exceed 500 mm/year. Given these challenging climatic conditions, establishing forests has not been a simple task, requiring a prolonged "trial and error" process [34]. Early pine monocultures proved to be vulnerable to pests, and in recent years were replaced with more diverse, indigenous assemblages [35,36]. A variety of water management innovations has allowed for forest stands to survive, at times in desert conditions. Israel's forestry efforts have been singled out for praise by United Nations officials and by experts responsible for combatting desertification as a model for other arid countries that seek to restore degraded lands [37].

Each of these three unique conditions, however, makes Israel's forests subject to criticisms. Even after they grow, the planted woodlands are often derided in some local ecological circles for their lack of aesthetic and ecological integrity [38]. While there appears to be considerable cross-cultural variability with regards to perceptions of forest aesthetics, certain forest characteristics (e.g., tree-stand factors) appear to influence perceptions of the attractiveness of a forest [39]. Hence, it should not be surprising that visits to relatively newly-planted forests have been disparaged as uninspiring and lacking the grandeur and mystery of long-established natural woodlands.

While disputed by many experts [40], some environmental activists and scientists also claim that Israel's forests contribute to the reduction in local biodiversity [41,42]. Moreover, as described above, because forests were used as part of the national strategy to ensure Jewish sovereignty over the land during the initial years of statehood, there has been a political critique of the forests. Opponents charge the KKL with exploiting afforestation to discriminate against the country's Arab minorities [43,44], even though such politicized forestry tactics have not been pursued for over fifty years.

And yet, forest visitation by the Israeli public is extremely high: Over a single, seven-day holiday period, some 2,000,000 people—or roughly 25% of the country's population (including both Jewish and Arab citizens)—were reported as visiting the local forests [45]. While annual visitation data at the national level are unreliable, a systematic study of the Ben-Shemen forest, a popular mid-sized woodland, located in the Tel Aviv–Jerusalem corridor, counted 1.5 million visitors in a single year [46]. As a densely populated country, there is a constant conflict between the forces of development and the environmental movement, which seeks to preserve Israel's open spaces [47]. Visiting the forests could be an effective tool for mobilizing public solidarity with environmental forces and preservation efforts. But it is not at all certain what effect these visits have on the Israeli public.

A 2002 study contrasted perceptions of KKL forests between visitors in the forest with those of the public who did not visit woodlands [48]. Over 300 visitors to KKL forests were interviewed, and their environmental attitudes were compared with telephone interviewees who had not visited a forest in over a year. Forests were identified as particularly popular destinations for families with young children, and 75% of the respondents in the forests were return visitors. Hiking and picnicking were ranked as the visitors' favorite activities associated with forest visits. The researchers found a statistically significant difference in the level of environmental commitment between the public that visited the forest and those who did not. This was measured by attitudes with regards to preservation/development dilemmas, donations to environmental groups, participation in environmental demonstrations, subscription to nature/environmental magazines, etc. Visitors also demonstrated a far higher environmental literacy

and ability to answer factual questions about ecological issues. But causality remained an open question. The research hypothesized, but could not prove, that visits to the KKL forest strengthened the environmental identity of the visitors. For example, researchers did not seek to sample a representative survey of Israel's population, but rather only sampled visitors in situ, at local forests. Accordingly, it is impossible to know whether the different major ethnic groups in the country had comparable or different experiences in the forests, as the secular Jewish majority was mostly represented in the survey.

Twenty years have transpired since this pilot research was conducted. The trees have grown, and Israeli society has evolved. Israel has become a more multi-cultural society, and its population density has increased by some 50%. These dramatic demographic changes raise important questions about the present experience of visits to forests and their impact on environmental opinions and commitments: Does the political and ecological critique of Israel's forests affect the diverse local citizens who regularly visit the forest or affect the way they perceive the local woodlands? In most developed countries, the public expresses considerable trust in its forestry administrations [49]. (This is particularly true in places such as Finland [50] or Switzerland [51].) Given the aforementioned controversies and political and ecological critique of Israeli foresters and local forest policy, do Israelis retain a positive attitude towards the KKL—Israel's national forestry agency—and the forests that it oversees?

## 3. Material and Methods

Random digit dial (RDD) telephone surveys in Israel face declining response rates and coverage. Accordingly, we chose to implement the survey instrument via a randomly recruited standing internet panel, supplemented with a randomly sampled telephone survey of non-panel members for a study of associations. The survey was conducted between March and April 2019. As one of the primary research objectives was to contrast different societal groups' visitation to forests and environmental perspectives, an emphasis was placed on ensuring a societally representative sample. The pre-recruited internet panel generated responses from 359 mainstream Jewish (secular/modern-Orthodox) Israelis residing across Israel, or 62% of the total sample size. Because Israel's Ultra-Orthodox and Arab citizens have been shown to be underrepresented in internet surveys, the resulting sample was supplemented by an additional 100 Ultra-Orthodox respondents (17.3%) and 120 Israeli Arab citizens (20.7%) who completed the survey instrument via telephone interviews. The combined pool of responses (579 respondents) offers an appropriate representation of these two significant population cohorts that represent 14 and 20 percent of local citizens, respectively. In addition, 49.7% of respondents were male and 50.3% female; 42% were age 18–34; 24% between the ages 35–44; 34% were age 45 and over. Some 82% of the respondents reported having children, and 12% were either single, widowed, or divorced.

The questionnaire was designed through an interactive process. The original draft was piloted by students who distributed the instrument via Facebook groups in July 2018. After receiving responses and feedback from 166 participants in the pilot run, significant revisions were made to the survey instrument. The final tested and revised questionnaire included some 30 substantive questions and a few additional questions to supplement the basic demographic information available about the internet panel participants. Online completion of the questionnaire and on the telephone averaged roughly fifteen minutes.

To tease out the role of the forests in strengthening such an environmental identity, five multivariate, regression analyses were conducted. Study participants were asked to respond to several pro-environmental statements, utilizing a Likert scale of 1–7, where 7 is "completely agree", 4 is largely neutral (don't agree–and don't disagree), and 1 was completely disagree. Continuous variables for the univariate analysis were compared by one-way analysis of variance test. Categorical variables were compared by the χ2 test. Statistical significance was set as two-sided at a minimum value of $p < 0.05$. Multivariate analysis for independent variables was also performed. Analysis was conducted by SPSS 25.0 package for Windows Software (SPSS Inc., Chicago, IL, USA).

## 4. Results

In general, visitation rates to Israeli forests were higher than anticipated: On average, respondents reported 1.2 visits per month to a KKL forest. We divided respondents into four levels of forest visitation frequency. Some 10% of respondents were classified as "intensive forest users" (with frequency reported to be one visit per week or more). On the other extreme, 15% of respondents were categorized as "occasional" with visits to the forest once a year or less. Figure 2 offers a breakdown of the forest visit frequency according to ethnic identity. Given the extensive literature which claims that Israeli Arab citizens are largely alienated from forests in Israel [52–54], it is surprising to find that Israeli Arab citizens reported relatively high rates of visits to KKL forests—with 2.2 visits on average per month, more than twice that of the mainstream Jewish (Jewish secular/modern Orthodox) population in Israel.

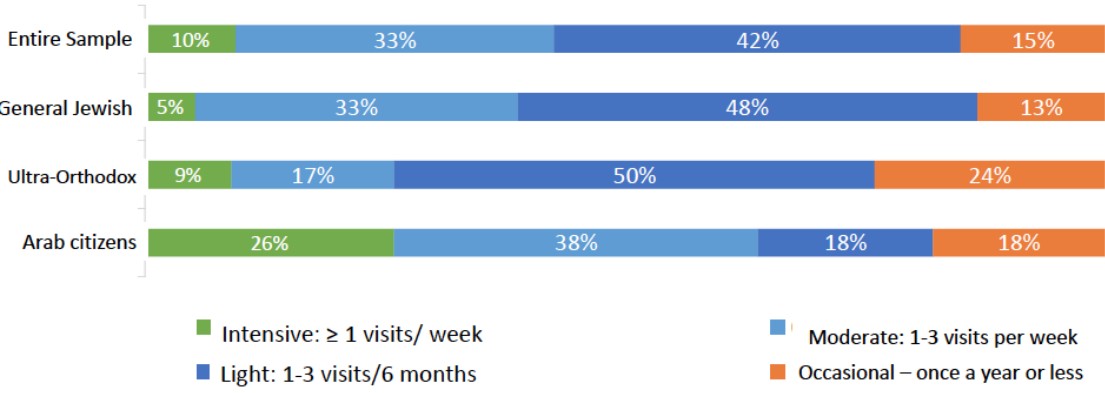

**Figure 2.** Frequency of visits by ethnicity.

This outcome may well be due to the fact that Arab municipalities tend to have far fewer parks and green spaces than do Jewish communities, for a variety of historical reasons. Israel's forests, therefore, offer a recreational refuge for families who lack local alternatives.

Respondents ranked a series of statements to assess environmental orientation via a 1–7 Likert scale. These included:

- It is important to protect the environment and nature for the public's benefit;
- It is important to invest in natural sites outside of cities;
- It is important to act on behalf of Israel's forests and the environment;
- I become excited when I see new forests in Israel, planted by the KKL; and
- It's important to invest in water resources because today, water resources are in trouble.

The influence of the following independent values was then considered for each statement:

- Age;
- Gender;
- Ethnic/religious affiliation;
- Geographic residence (south, central, northern Israel).
- Level of education (with five discreet levels; elementary, secondary, post-high school without a degree, an undergraduate degree, and graduate-level training.
- Frequency of visits to forests (four levels as described above).

The first and perhaps most important statement to which respondents offered a personal ranking was, "It is important to protect the environment and nature in Israel." We conducted a "Stepwise" analysis (R = 0.228) with an explained variation of 0.049. Among the variables that appeared to be relevant in influencing stated commitment to protecting nature and the environment involved "ethnic affiliation". Respondents were divided into three groups: *Secular and modern Orthodox Jewish Citizens*,

*Ultra-Orthodox Jewish Citizens,* and *Arab Israeli citizens.* Another demographic trait evaluated was respondents' geographical location (*North/Haifa region; Central/Jerusalem Region/South and Coastal Region*). Ethnic identity was found to be the more influential independent variable in predicting respondents' opinions (Absolute beta indicators 0.172); Geographic identity (0.117) had a more modest influence. Table 1 offers a summary of the relevant associations in the aforementioned regression model.

**Table 1.** Factors affecting commitment to the protection of the environment and open spaces.

| Model | R | R Square | Adjusted R Square | Model Summary Std. Error of the Estimate | Change Statistics | | | | |
|---|---|---|---|---|---|---|---|---|---|
| | | | | | R Square Change | F Change | df1 | df2 | Sig. F Change |
| 2 | 0.228 | 0.052 | 0.049 | 1.01643 | 0.013 | 7.664 | 1 | 554 | 0.006 |

**Coefficients**

| Model | | Unstandardized Coefficients | | Standardized Coefficients | t | Sig. |
|---|---|---|---|---|---|---|
| | | B | Std. Error | Beta | | |
| 2 | (Constant) | 6.451 | 0.165 | | 39.007 | 0 |
| | Ethnic Affiliation | 0.235 | 0.058 | 0.172 | 4.064 | 0 |
| | Geographical Region | −0.179 | 0.065 | −0.117 | −2.768 | 0.006 |

The second regression model assessed the responses to the statement: "It is important to invest in natural sites outside of the cities" in a matter comparable to the previous regression equation. Here, the Stepwise analysis yielded an $R^2$, association of 0.093, and an explained variation of 0.007. The only variable which proved to be important was the frequency of visits to the forest. In other words, the more a respondent spends time in forests, the greater the preference for investment in non-urban natural sanctuaries. Specific results of the regression model are presented in Table 2.

**Table 2.** Factors affecting commitment to investment in forests outside urban areas.

| Model Summary | R | R Square | Adjusted R Square | Model Summary Std. Error of the Estimate | Change Statistics | | | | |
|---|---|---|---|---|---|---|---|---|---|
| | | | | | R Square Change | F Change | df1 | df2 | Sig. F Change |
| 1 | 0.093 | 0.009 | 0.007 | 1.35635 | 0.009 | 4.747 | 1 | 549 | 0.03 |

**Coefficients**

| Model | | Unstandardized Coefficients | | Standardized Coefficients | t | Sig. |
|---|---|---|---|---|---|---|
| | | B | Std. Error | Beta | | |
| 1 | (Constant) | 5.777 | 0.176 | | 32.791 | 0 |
| | Frequency of Forest Visitation | 0.152 | 0.07 | 0.093 | 2.179 | 0.03 |

The components of an "environmental identity" are complex and frequently dependent on an individual's ecological reality and available natural resources. It appears that closeness to nature and concern for natural resources are not synonymous. For instance, one statement to which survey participants responded was, "It is important to save water and invest in the country's water system because today the water system faces a crisis". Responses again could range from 1 to 7. Using the Stepwise analysis, R was equal to 0.145, and explained variation was 0.019. Here, frequency of forest visits proved to have only a modest association with responses. At the same time, ethnic affiliation emerged as the only demographic variable that informed opinions: Arab Israelis tended to offer a more positive evaluation of this statement, strongly identifying with the need to conserve water and invest in water infrastructure while reflecting a stronger commitment to these aims than Jewish Israelis.

The regression analysis includes a statement: "KKL (Israel's forestry corporation) needs to be an actor in matters of forestry and environment". Here the R was equal to 0.139, with the explained variation 0.017 with age having the most powerful association: older Israelis tended to be more positive about this organizational priority. Another statement involving forests held "I am moved when I see

the new forests that the KKL has planted on behalf of the citizens of the state". Here the R = was only 0.307 and explained variation 0.087. The two most influential variables were age and visitation frequency, 0.227 and 0.155, respectively, with gender and education having only a modest impact. See Table 3.

**Table 3.** Factors affecting the emotional impact of forests on respondents.

| Model | R | R Square | Adjusted R Square | Model Summary Std. Error of the Estimate | Change Statistics | | | | |
|---|---|---|---|---|---|---|---|---|---|
| | | | | | R Square Change | F Change | df1 | df2 | Sig. F Change |
| 4 | 0.307 | 0.094 | 0.087 | 1.73085 | 0.01 | 5.831 | 1 | 525 | 0.016 |

**Coefficients**

| Model | Unstandardized Coefficients | | Standardized Coefficients | t | Sig. |
|---|---|---|---|---|---|
| | B | Std. Error | Beta | | |

Another important finding involves the generally strong support for environmental positions among Israelis. This is deduced by their placement on an environmental ethics continuum (Figure 3), which runs between two polar opposite positions—one reflecting strong environmental concerns and the other an alternative value. Using a Likert Scale, respondents were situated along the continuum, indicating their relative commitment to competing environmental positions and values. Figure 3 shows the percent gap between those who strongly support a given statement (ranking it 6–7 where maximum possible support was 7) and those who prefer a stance located on the opposite end of the ethical spectrum.

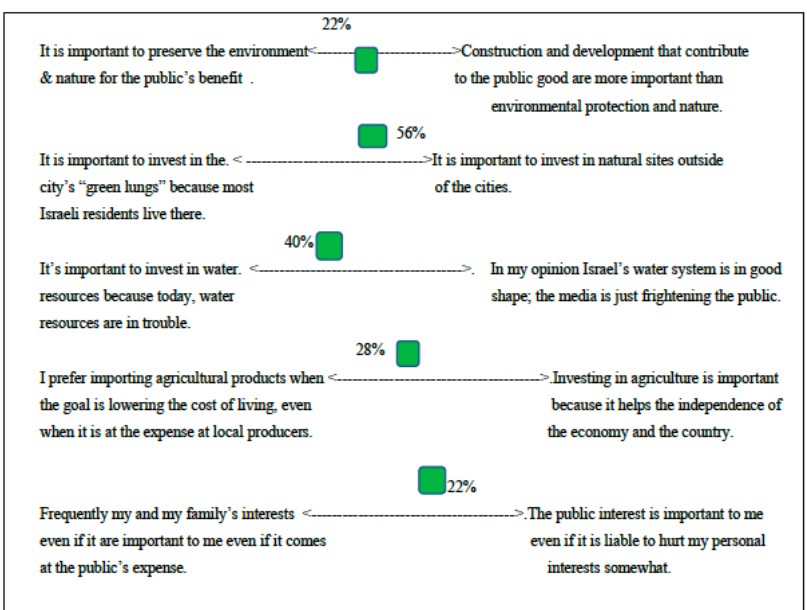

**Figure 3.** Environmental values clarification—general public.

Accordingly, the following five statements offer a snapshot of general attitudes held by the Israeli public, allowing respondents to identify where they stand with regards to conflicting positions dividing the public over local environmental controversies. The percentage points in the figure indicate the relative preference of respondents for a given statement on the ethical continuum. (e.g., 22% more respondents generally prefer nature protection to development).

For instance, when two competing statements are set head-to-head at opposing poles, some 22% *more* Israelis express a general commitment to preservation values rather than to development. One unexpected finding was the strong preference for nature conservation activities in the country's

less accessible, peripherally-located open spaces than in urban areas. For example, 56% more of the sample believe that conservation investments should be prioritized in the open spaces outside of cities, rather than in the urban areas. In general, there is an extremely high level of concern for Israel's water resources and a belief that they are in trouble; 40% more Israelis perceive the local water system as needing help than those who are less troubled by the issue.

A surprising result was the low general level of concern for "food security" and sense of solidarity among citizens for Israel's agricultural producers. The overwhelming preference for increased importation of food to lower the prices (28% greater support was expressed for this view) is inconsistent with the popular view of preference for local products. It also flies in the face of the rising concerns in environmental circles about the consequences of the country's shrinking cultivated farmlands and the associated perils of reduced local calorie production [55]. Israel, in general, also emerges as a very public-minded society, at least in their public pronouncements. A high percentage of respondents *express* a willingness to forego *individual* interests in favor of *public* interests when they conflict. Specific results about societal attitudes on these continua are displayed in Figure 3.

These findings offer an important "baseline" for the central question of the present study, What impact do visits to forests have on the environmental commitments of visitors in their day-to-day life, after they leave the natural and the recreational context? Are frequent visits to forests associated with a "greener" perspective when faced with the dilemmas associated with environmental issues? Further analysis reveals an association between respondents who have visited the forest with greater frequency and environmentally favorable positions.

In the environmental ethics continuum presented above, frequent visitors were far more likely to express commitment to conservation than development values. Indeed, while all groups in Israel tend to generally support "preservation" over "development", such views are far stronger (17% higher) among frequent than infrequent forest visitors. Concern for Israel's water resources, in general, was extremely high, presumably because the survey was taken after five consecutive drought years. Even so, frequent forest visitors still showed somewhat more concern (5% more) than the less frequent or non-visitors. There was also a 9% greater inclination to support agriculture and Israel's food security and independence among frequent visitors than non-visitors. The latter strongly preferred import policies, which reduced food prices, which reduced support for local farmers. (Nonetheless, a majority of frequent visitors (59%) leaned towards policies that increase exports and reduce support for farmers.)

Surprisingly, in predicting a preference for investing in nature preservation in the non-urban open spaces versus cities, forest visitation per se does not appear to be an influential factor. Both frequent and infrequent forest visitors support this as a policy priority. It is also interesting to note that a stronger majority of Israelis who visit forests report a greater willingness to sacrifice personal interests in favor of public interests than those who do not (8% more than infrequent visitors).

In another section of the questionnaire, over half (55%) of Israelis expressed identification with the statement: "I am excited when I see the new forests planted by the KKL on behalf of the residents of the state". But not surprisingly, this position was far more highly associated with frequent visitors. (Pearson correlation of 0.143—a correlation significant at the 0.01 level (2-tailed).) At the same time, it is important to note that identification with such a statement was made without having to consider a competing, and possibly compelling, alternative view.

It is also worth noting different nuances among Israel's different ethnic groups. Specifically, 88.5% of Ultra-Orthodox respondents supported the statement that it was important to protect the environment and nature for the public's benefit. This is significantly greater than the 73.5% among non-frequent or non-visitors. Similarly, frequent Ultra-Orthodox visitors were more likely to support investing in forests and natural sites outside cities (96% of respondents) than infrequent visitors (90%)—even as this level of commitment was already quite high. Little difference was found among the Ultra-Orthodox and the general public's frequent forest visitors in matters involving individual support for conservation and development of water resources.

Among Arab-Israeli citizens, the results were somewhat different. Frequent Arab visitors were more prone to support investment in forests and recreation areas outside of the urban areas (87% versus 78%). But no special expression of environmental values or to water conservation emerged.

## 5. Discussion

Environmental activists have long been concerned about the general lack of commitment to environmental practices and the prevailing unwillingness to sacrifice for common ecological aims among the general public around the world [56]. Presumably, without greater individual engagement, little progress will be made in addressing critical ecological challenges (e.g., climate change, biodiversity preservation). Increasing the sense of responsibility for ecosystems requires a strengthening of the public's ideological and visceral commitment to the natural world. As pressures on open spaces and natural sites mount due to population growth, bolstering public commitment to the preservation of these forested places becomes increasingly important if they are to be conserved. As in most Western societies, Israelis increasingly adopt urbanized lifestyles in which virtual reality and computer screens steadily replace direct human interactions with the natural world. Under such circumstances, saving open spaces and woodlands for the future emerges as an even greater challenge and paramount priority for governmental and non-governmental conservation agencies and advocates.

Considerable focus has been placed on the psychological barriers to pro-environmental behavior [57] and the socioeconomic factors affecting people's ecological perspective [58]. This would suggest that improving environmental behavior requires broader, long-term interventions targeting social gaps and economic opportunity. There is much evidence that suggests that the public, from all socioeconomic niches, cares about the natural world and can be influenced to seek greater harmony with the environment. Much research focuses on the effectiveness of sending subtle and not-so-subtle messages to nudge the general public towards greater ecological responsibility [59]. Many experts posit that there should be a more significant spiritual component in the promotion of environmental behavior [60].

The academic literature that focuses on the alienation of Israeli Arab communities to the land of Israel [61,62], along with the bitterness expressed by the community's political representatives, produces a commonly-held notion that assumes pervasive resentment towards Israel's forest among Arab citizens. This may be an outdated misconception. We find that such previous claims were not based on empirical data. In the present study, the first systematic and replicable survey of its kind, Israeli Arabs consistently report positive views about their forest experiences—views similar to those of Jewish respondents. They emerge as particularly frequent "woodland consumers", extremely enthusiastic about the Israeli forest experience and an engaged minority community.

We also found practically no meaningful signs of ecological or aesthetic dissatisfaction with the state of Israeli forests among the general public. This suggests that ecological critiques of Israel's forests simply do not resonate with the general public. The results are important for land managers and environmentalists in areas considering dryland restoration through afforestation strategies. The results are also important for advocates of greater cooperation and improved coexistence between Israel's Jewish and Arab citizens. The potential effect of restored woodlands appears to go far beyond simple recreation. Rather, newly-planted, dryland forests still offer a profound experience to visitors and can inspire support for forests and afforestation initiatives.

We argue that as the people on the planet increasingly make their homes in cities and suburbs, visits to forests, even recently planted ones, remain powerful forces that influence human perspectives about environmental issues. Expanding access to woodlands has become a more important factor in advancing support for nature conservation than ever before. Beyond the aforementioned psychological and physiological benefits provided by regular access to woodlands, visits to forests increase appreciation for these natural resources and affect individual support for protecting them. Our findings suggest that in Israel, this effect appears to cross across ethnic boundaries. Indeed, the environmental commitment of Israel's Arab community (which is typically less affluent than the Jewish majority) may be more

influenced by contact with the country's forests than the Jewish public. That is because Arab citizens not only visit Israeli forests more frequently than their Jewish counterparts but also feel more strongly, as a result, that it is important to "protect the environment and nature in Israel". Israel's Ultra-Orthodox minority also holds strong, pro-conservation perspectives.

A favorable tendency, identifying with the environment and nature, exists across Israel's highly heterogeneous population. Getting people of all ages out of their homes and neighborhoods and into the forests appears to be an important way to strengthen this impulse. Clearly, causation is very difficult to "tease out" in such analyses; individuals visiting forests are clearly predisposed to having positive feelings towards nature. However, the high association found in our study with a range of environmental impulses and commitments to public wellbeing suggests that frequent outings and simply spending time in forests helps maintain these qualities.

Newer, human-planted forest stands, such as those characterizing most of Israel's woodlands, may seem less diverse and inspiring to ecological purists than biologically diverse, old-growth woodlands [63]. Nonetheless, most Israelis (including over half of Arab respondents) express excitement at seeing Israel's newly established forests.

For some time, the important contribution of afforestation to global strategies for addressing the climate crisis has been well accepted [64]. New dryland stands have been measured with surprisingly high levels of carbon sequestration [65]. While tree planting should not be the "end all" of climate policy, it is an important component [66].

It is well to remember that expanding forest cover and planting trees poses opportunity costs. Public support for these activities is critical. Getting people to forests (or presumably to other natural sites) appears to be a simple and effective way to foster appreciation of natural systems and the ancillary benefits associated with afforestation. It also serves to bolster public solidarity with the environment and open space preservation in general.

## 6. Conclusions

Israel is an extreme example of an urban society; less than ten percent of the public lives in rural settings. This makes the connection between humans and forests particularly powerful experiences. As environmental educators and advocates consider strategies for maintaining the country's generally high level of commitment to conservation and environmental protection, forest visitation should be part of the package. Dryland nations considering afforestation initiatives should not feel that their local woodlands are inferior. This study suggests that the experience of the public who comes in contact with these new forests can be no less profound than visits in more temperate lands.

Our findings provide clear answers to the research questions, suggesting that the more frequently people visit the forests in Israel, the stronger is their commitment to the environment. Rather than finding reduced participation among Arab Israeli citizens, our studies suggest that they visit forests more frequently and are generally more positive about the KKL forests than Jewish Israelis. Ultra-Orthodox Israelis who visit forests often also emerge as highly supportive of environmental protection. Further qualitative research is important, not only to confirm the causality suggested in this study but to characterize the nature of the experience for visitors. Policymakers and foresters should consider how time spent in the forest can be leveraged to upgrade the educational and inspirational impact.

In Israel, the population is growing rapidly, at an annual rate of 2% [67]. As cities become even more crowded, urban residents will increasingly seek the refuge of the countryside in shaded open spaces for recreation, communion with nature, and a modicum of contemplative solitude. While there is room for modest expansion of Israel's forests, most of the land that can be zoned to that end has already been planted. Accordingly, to meet the growing demand for woodlands and parks, a steady, annual investment is critical to develop the additional hiking and cycling trails, picnic tables, recreational facilities and even parking for the future.

Decisionmakers should also be aware that large segments of the Israeli public do not own a car [68], making many local forests essentially inaccessible for them. Typically, there is an association between

socioeconomic level and vehicle ownership. This suggests that leaving Israeli woodlands as a resource only available to car owners exacerbates existing social and environmental injustice. Increasing the public's accessibility to the forests (by influencing the frequency, reliability, and convenience of public transport to forests and parks) will serve to increase the number of citizens who can take pleasure in recreation and enjoy the general edification offered by the country's unique woodlands. Based on the findings of the present research, there is a solid basis for assuming that this will also generate ecological and civic dividends by strengthening citizens' environmental ethos and commitment to conservation.

**Author Contributions:** A.T. designed the first draft of the survey and wrote the original draft of the article. M.B. revised the survey, oversaw the statistical analysis, and edited the final article. All authors have read and agreed to the published version of the manuscript.

**Funding:** This research received no external funding.

**Acknowledgments:** The authors express their gratitude for support from the KKL Institute for Zionism and Settlement and the Eastern R&D Center, Israel. Retired from Technion University, Yonina Rosenthal provided valuable comments to an earlier draft.

**Conflicts of Interest:** The authors declare no conflict of interest.

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
