# Peer review of "The Impact of Visits to Dryland Forests on Environmental Outlook: Results from a National Survey"

_forests, doi:10.3390/f11080872_

Round 1
Reviewer 1 Report
This is an interesting paper, particularly for those who are not familiar with the region and its issues. One suggestion is to provide more background on forest resources in Israel. Perhaps a map of the country with its forest resources can be provided, perhaps with major population centers, as determined relevant for the paper.
It seems that the last to paragraphs of the Introduction really belong to the Discussion or Conclusion sections.
What is the reason that the regression results are not provided in the paper? It would help make the paper more informative and also evaluate the results.
Perhaps in the last two sections the authors can offer more of their own views on what explains their results and what would be the policy implications.
The manuscript needs some language editing when it comes to word choice, syntax, and grammar.
Author Response
Response to Reviewers’ Comments: Forests Manuscript Number: 884208
The Impact of Visits to Dryland Forests on Environmental Outlook: Results from a National Survey
(Alon Tal & Miriam Billig)
Reviewer 1:
Comment 1: This is an interesting paper, particularly for those who are not familiar with the region and its issues. One suggestion is to provide more background on forest resources in Israel. Perhaps a map of the country with its forest resources can be provided, perhaps with major population centers, as determined relevant for the paper.
Response: This is an excellent idea. We include a new section 2 with a brief description of local forestry history as can be seen in lines 65-144. We have also added a new figure 1, which as suggested, contains two maps contrasting the forests of Israel during the first half of the twentieth-century with the present forest cover, reflecting decades of significant afforestation work. We believe that this fully responds to the suggestion.
Comment 2:It seems that the last to paragraphs of the Introduction really belong to the Discussion or Conclusion sections.
Response: We agree. These two paragraphs have been moved to the Discussion (bottom of p. 12) as suggested.
Comment 3: What is the reason that the regression results are not provided in the paper? It would help make the paper more informative and also evaluate the results.
Response: We agree. The new tables added to pages contains all the relevant information about the regression analysis. (See pages 8-9.) We trust that your graphics people can harmonize these excel tables with the article text as we find it in the formatted version.
Comment 4: Perhaps in the last two sections the authors can offer more of their own views on what explains their results and what would be the policy implications.
Response: We agree. We have added the equivalent of two paragraphs about the policy implications of our findings to the concluding section. (See lines 437-450.)
Comment 5:The manuscript needs some language editing when it comes to word choice, syntax, and grammar.
Response: We agree. We have had the article reviewed by two native-English speakers, including a retired professor of English. We believe that the language level is now sufficient for international academic journal standards.
Reviewer 2 Report
Review Report – Forests Journal Article
Review Report
- A brief summary
The paper highlighted the importance of urban forest in Israel for as a space for human-forest interaction, quoting the fact that only 10% of the Israeli population are living in rural areas, the unique history of the urban forest, and the meaningful role of Israeli’s forest in local political dynamics. By conducting a national survey with a total of 579 respondents, the study evaluated the impact of forest visitation to the perspectives and attitudes of the visitors towards local forest. The study found that there are no meaningful signs of ecological or aesthetic dissatisfaction from visitors. Different nuances among different ethnic groups were identified. The findings also suggest that the frequency of people’s visitation to the forests is correlated positively to their commitment to the environment. The authors concluded by making some recommendations for policymakers and foresters in order to advance with the educational and inspirational impact of forest visitation.
- Broad comments
Strength
- The paper discusses issues which are very relevant to the current situation and can benefit the practitioner as well as policymakers in the respective area, moreover in light of the COVID19 pandemic which put human-forest interaction at the forefront of nature-based solutions.
- The study concluded by presenting clear and specific argumentation and recommendation on how policymakers and city authority can take the issue to the next level, inter alia through enhanced public education on environmental issues, and improved transportation infrastructure to support visit to the forest,
- The findings of this study bring new insights to the issues of interest, and provides room for further study and discussion, particularly on the relevance of ethnicity towards attitude to forest protection.
Potential areas for improvement
- The introduction:
- Given the varying definition of forest in a different context, it would be useful if authors can define ‘forests’ in the context of this study in the first place. That way, readers can get a better understanding of the actual context of ‘visitation to forests’ which was referred to by the authors -- as the central issues of the study.
- Given the nature of introductory section, the two last paragraphs of ‘Introduction’ I think would be better to be placed elsewhere – maybe results or discussions – in order to keep the reading excitement and to avoid repetition.
- Research questions can be made sharper, clearer and should be consistently referred to throughout the text.
- The third last paragraph I think has been somehow expressed in the earlier paragraph. Perhaps lines 126-127 can be blended with lines 118-119. Thus, avoiding repetition. Additionally, an example of inconsistencies for the research question: lines 286-290 mentioned another set of questions which, although of similar interest, are slightly different from those posed in the introduction part. Having a more consistent research question I think would be helpful to help the reader identify whether or not that questions would have been answered in the results/discussions sections.
- The methods:
- It would be useful to explain a bit on the reasoning behind the number of samples used. A hint of the population number will be also useful to get the feel on the magnitude of the sample. The argumentation on selecting three ethnic groups (Jewish, orthodox, Arab), in terms of their representativeness towards the whole Israeli’s population can be made clearer.
- How the data analyzed can be briefly added in this section to make this section more informative and transparent. For example, the statistical analysis
- Some parts of the results (for example, line 178 – 202) I think should be placed in the ‘methods’ section, given that they are explaining how to get to the results, not the results itself. The results part should then directly mention the resulting analysis or resulting figure once all the steps in the methods sections were undertaken.
- Results:
- as mentioned above, I believe the results section should no longer explain the ‘how to’ get to the resulting figure. The ‘how-to’ explanations should be placed in the method section.
- It would be useful if Figure 2 can be reconstructed to better represent and provide the preview/highlight to that specific results. As per now, the figure (which mostly are text) is not easily read – the continuum is not clearly shown. Also, the title of Figure 2 should be shortened; more detail explanation can be put in the text.
- I also noted the inconsistency between method and results on the categorization of respondents, where line 150 (methods) mention ‘359 Israelis’, while Figure 1 uses the term ‘general Jewish’. If this refers to similar sub-population being sampled, it would be useful to use a consistent term of categorization throughout the article.
- Discussions
- It would be interesting if the authors can briefly hint whether or not other independent values also factor in the resulting output, as mentioned in line 189 – 194 (age, gender, level of education).
- Conclusions
- I think it would be really helpful to the reader if the answer to the specific research questions
- Specific comments
- Title: instead of ‘man-made’ forest, perhaps a more gender-neutral terminology can be used, such as ‘human-made’ forest or ‘plantation forest’. It is also interesting to note that the term ‘man-made forest’ only appear within the title, and not within the article.
- Introduction: line 36, (‘Bringing school children..), there are two ‘has been…”
- Line 208 – 209: “In other words, the more frequent a respondent forests, …….” , I think something missing in that sentence. What does it mean by ‘the more frequent a respondent forest?’
- Line 242 – “56% of the public” or “56% of the sample ?”
- Line 169-170 “given the extensive literature…” maybe the authors can add references; some examples of this extensive literature.
Author Response
Response to Reviewers’ Comments: Forests Manuscript Number: 884208
The Impact of Visits to Dryland Forests on Environmental Outlook: Results from a National Survey
(Alon Tal & Miriam Billig)
Reviewer 2:
Potential areas for improvement
The introduction:
- Comment 1: Given the varying definition of forest in a different context, it would be useful if authors can define ‘forests’ in the context of this study in the first place. That way, readers can get a better understanding of the actual context of ‘visitation to forests’ which was referred to by the authors -- as the central issues of the study.
Response: We agree. On line 68, we now offered a translated definition of forests provided in the definitive Hebrew dictionary, as an additional indication of how local forests might different from conventional woodlands.
Comment 2: Given the nature of introductory section, the two last paragraphs of ‘Introduction’ I think would be better to be placed elsewhere – maybe results or discussions – in order to keep the reading excitement and to avoid repetition.
Response: We agree. Reviewer 1 made the same suggestion and these paragraphs have been moved to the Discussion section as described above.
Comment 3: Research questions can be made sharper, clearer and should be consistently referred to throughout the text.
Response: We agree. The two central research questions are fully stated in lines 53-57.
Comment 4: The third last paragraph I think has been somehow expressed in the earlier paragraph. Perhaps lines 126-127 can be blended with lines 118-119. Thus, avoiding repetition. Additionally, an example of inconsistencies for the research question: lines 286-290 mentioned another set of questions which, although of similar interest, are slightly different from those posed in the introduction part. Having a more consistent research question I think would be helpful to help the reader identify whether or not that questions would have been answered in the results/discussions sections.
Response: We generally agree. Accordingly lines 126-127 have been changed so that they are now largely different than lines 118-119 and serve as more of a transition to the subsequent paragraphs in the article.
With regards for the rest the questions posed in lines 286-290, we believe that restating the questions helps readers to stay focused on the research questions and the theoretical presentation that appears at the opening of the article. Merely repeating the research questions would in our opinion become repetitive for readers. But we feel that the present presentation is clear and keeps readers on track. The entire paragraph can be removed if the editor feels otherwise. But our inclination is to leave it in its present form.
The methods
Comment 5: It would be useful to explain a bit on the reasoning behind the number of samples used. A hint of the population number will be also useful to get the feel on the magnitude of the sample. The argumentation on selecting three ethnic groups (Jewish, orthodox, Arab), in terms of their representativeness towards the whole Israeli’s population can be made clearer.
Response: We agree. We have now added a sentence about the rationale behind intervening to ensure a representative sample and offered more information about the different demographic breakdowns among respondents. (See lines 179-189.)
Comment 6: How the data analyzed can be briefly added in this section to make this section more informative and transparent. For example, the statistical analysis.
Comment 7: Some parts of the results (for example, line 178 – 202) I think should be placed in the ‘methods’ section, given that they are explaining how to get to the results, not the results itself. The results part should then directly mention the resulting analysis or resulting figure once all the steps in the methods sections were undertaken.
Comment 8: as mentioned above, I believe the results section should no longer explain the ‘how to’ get to the resulting figure. The ‘how-to’ explanations should be placed in the method section.
Response 6-8: We merged our response to reviewer 2 comments 6 through 9 as they address the same point, with comment 7 and 8 simply suggesting a way to better explain description of the statistical analysis and shift the place of a few sentences. In response, we have added a full, additional paragraph to the methods section which describes the data analysis, statistical tests used, significance level constraints, software package, etc. (Lines 195-202.) We prefer not to list the actual questions from the survey in the methods as we believe that readers will need to have them repeated soon thereafter in the results section to understand the findings, creating duplication. But we believe that the additional description probably meet the expectation of Reviewer 2 in this regard.
Results:
Comment 9: It would be useful if Figure 2 can be reconstructed to better represent and provide the preview/highlight to that specific results. As per now, the figure (which mostly are text) is not easily read – the continuum is not clearly shown. Also, the title of Figure 2 should be shortened; more detail explanation can be put in the text.
Response: We agree with the general critique and have shortened the title of the Figure 3 (previously Figure 2, before adding the map requested by Reviewer 1). Additional language was added to the text leading up to figure that describes the logic and design of the graphical presentation and we believe it is now clearer to readers. We fear that part of the confusion might be a result of changes in the word file caused by the font change which distorted the original figure. We have exchanged the original word figure with a figure based on a pdf box to prevent distortions during the present review. We will pay special attention to proofs to make sure that the figure that is published precisely represents the findings.
Comment 10:I also noted the inconsistency between method and results on the categorization of respondents, where line 150 (methods) mention ‘359 Israelis’, while Figure 1 uses the term ‘general Jewish’. If this refers to similar sub-population being sampled, it would be useful to use a consistent term of categorization throughout the article.
Response: This is a good point. We have changed the wording so that it precisely reflects the language used in the demographic breakdown presented in the methods.
- Discussions
Comment 11: It would be interesting if the authors can briefly hint whether or not other independent values also factor in the resulting output, as mentioned in line 189 – 194 (age, gender, level of education).
Response: The aforementioned inclusion of the regression tables, as requested by Reviewer 1,which include all independent variables showing significant influence on the dependent variable addresses this suggestion. We have also highlighted a few of the more interesting associations as suggested.
Conclusions
Comments 12: I think it would be really helpful to the reader if the answer to the specific research questions.
Response: Done. (See lines 432-435.)
Specific comments
- Comment 13: Title:instead of ‘man-made’ forest, perhaps a more gender-neutral terminology can be used, such as ‘human-made’ forest or ‘plantation forest’. It is also interesting to note that the term ‘man-made forest’ only appear within the title, and not within the article.
Response: This is a good point. We believe that more salient to the research than their being planted by humans, is the fact that the forest are located in the “drylands”, which makes them very different than temperate or tropical forests. So we have removed “man-made” from the title and replaced it with “dryland” which we believe is more germane in presenting the study.
- Comment 14: Introduction:line 36, (‘Bringing school children..), there are two ‘has been…”
Response: Corrected. Thank you.
- Comment 15: Line 208 – 209: “In other words, the more frequent a respondent forests, …….” , I think something missing in that sentence. What does it mean by ‘the more frequent a respondent forest?’
Response: Right again. The corrected sentence now reads: “In other words, the more a respondent spends time in forests, the greater the preference for investment in non-urban natural sanctuaries .”
Comment 16: Line 242 – “56% of the public” or “56% of the sample ?”
Response: Yes. Corrected.
- Comment 17: Line 169-170“given the extensive literature…” à maybe the authors can add references; some examples of this extensive literature.
Response: Three more examples of academic articles/book chapters devoted to this subject are now footnoted, in addition to the three sources cited earlier.